# Effects of Various Nectar and Pollen Plants on the Survival, Reproduction, and Predation of *Neoseiulus bicaudus*

**DOI:** 10.3390/insects15030190

**Published:** 2024-03-13

**Authors:** Yue Han, Wurigemu Lipeizhong, Xinqi Liang, Zhiping Cai, Weiru Liu, Jifei Dou, Yanhui Lu, Jianping Zhang, Shaoshan Wang, Jie Su

**Affiliations:** 1Key Laboratory of Oasis Agricultural Pest Management and Plant Protection Resources Utilization, College of Agriculture, Shihezi University, Shihezi 832003, China; hanyue1@stu.shzu.edu.cn (Y.H.);; 2State Key Laboratory for Biology of Plant Diseases and Insect Pests, Institute of Plant Protection, Chinese Academy of Agricultural Sciences, Beijing 100193, China

**Keywords:** Phytoseiidae, nectar and pollen plants, biological control, fecundity, predation

## Abstract

**Simple Summary:**

Floral resources have been increasingly utilized in conservation-based biological control to support the natural enemies of insect pests or pest mites. The native predatory mite *Neoseiulus bicaudus*, found in Xinjiang, plays a crucial role in the integrated pest management of agricultural crops. The aim of this experiment was to compare the impacts of floral resources on the longevity, fecundity, and predation ability of *N. bicaudus*. Finally, a comparative analysis of the adaptation of eight nectar and pollen plants to *N. bicaudus* was performed, as well as further investigation into the effect of pollen from these nectar and pollen plants on the predation ability of predatory mites. The obtained results have significant implications for the utilization of nectar and pollen plants for eco-friendly pest mite control in farmland.

**Abstract:**

*Neoseiulus bicaudus* is a predatory mite species that could potentially be used for the biological control of spider mites and thrips. Floral resources can provide excellent habitats and abundant nutrients for natural enemies. The objective of this experiment was to evaluate the effects of eight floral resources on the longevity, fecundity, and predation ability of *N. bicaudus*. Among the considered plants, *Cnidium monnieri* led to the highest longevity (24 days) and fecundity (13.8 eggs) of *N. bicaudus*, while *Tagetes erecta* resulted in the lowest longevity (7 days) and fecundity (0.1 eggs) observed in the predatory mites. By comparing the effects of three nectar and pollen plants on the predation of predatory mites, it was observed that *N. bicaudus* still exhibited a type II functional response to *Tetranychus turkestani*. In the presence of pollen, the predation efficacy (*a*/*T_h_*) of *N. bicaudus* exhibited a lower value, compared to that in the absence of pollen (Control: *a*/*T_h_* = 24.00). When pollen was supplied, the maximum consumption (1/*T_h_*) of predatory mites was higher than in its absence (Control: 1/*T_h_* = 9.90 d^−1^), with the highest value obtained in the presence of *B. officinalis* pollen (*B. officinalis*: 1/*T_h_* = 17.86 d^−1^). The influence coefficient of predation of *N. bicaudus* on *T. turkestani* in the presence of pollen was compared in the presence of three nectar and pollen plants: *Cnidium monnieri*, *Centaurea cyanus*, and *Borago officinalis*. At low prey densities, the influence coefficient of *C. cyanus* exceeded that of *B. officinalis*, and the overall influence coefficient values were negative (i.e., the presence of pollen reduced predatory mite feeding on *T. turkestani*). They exhibited similar values at high prey densities, and all of the influence coefficient values were close to 0 (i.e., the presence of pollen had no effect on predatory mite feeding on *T. turkestani*). The findings revealed that diverse plant species exert differential impacts on *N. bicaudus*, with some influencing its lifespan and others affecting its reproductive capabilities. Furthermore, the presence of nectar and pollen plants had a significant impact on predatory mite feeding on *T. turkestani* at low prey densities; however, this effect diminished as the prey density increased. Therefore, we recommend planting *C. monnieri*, *C. cyanus*, and *B. officinalis* in the field to ensure an ample population of predatory mites. The obtained results hold significant implications for the utilization of nectar and pollen plants in eco-friendly pest management strategies within agricultural contexts.

## 1. Introduction

In agro-ecosystems, predatory and parasitic natural enemies are a valuable biological control resource for major pests; however, optimizing their effective control often requires additional resources [1]. The “Predator-in-First” (PIF) strategy aims to enhance the abundance of natural enemies by capitalizing on their broad dietary preferences and ability to sustain and reproduce using the resources provided by plants in the absence of pest organisms. This strategy can be considered as an agro-ecosystem management technique that utilizes functionally attractive plants to serve as resource reservoirs supporting natural enemies [2,3,4]. Functional plants refer to a category of plant species that possesses the ability to perform essential ecological functions within agricultural and forestry ecosystems. In recent years, nectar and pollen plants—as a specific type of functional plant—have garnered significant attention and interest [5,6]. These plants may serve as a source of pollen and nectar for prey, while simultaneously providing alternative food and shelter for natural enemy insects or pollinators [7].

In nature, there exists a diverse array of over 200,000 species of flowering plants, which serve as exceptional habitats and sources of abundant nutrients for natural enemies [8]. In situations where the targeted prey is absent or scarce, these natural enemies can sustain themselves by consuming pollen or nectar from alternative plant sources, which are frequently limited in agricultural systems [9]. The presence of flowering plants ensures the availability of essential resources to sustain natural enemy populations throughout the entire duration of the season [10]. Parasitoids have been documented to derive benefits from a diverse range of nectar and pollen plants, such as those in the Polygonaceae, Umbelliferae, Cruciferae, Boraginaceae, Solanaceae, and Rosaceae families. These plant species are frequently selected as nectar resources in field studies, due to their ability to extend the lifespan of parasitoids [11]. Furthermore, studies on the effects of integrated traits have revealed that plants exhibiting a compound umbel or raceme inflorescence form and shallow corolla have the highest impact on parasitoid longevity [12]. Similar to parasitoids, hoverflies are strictly dependent on floral resources during their adult stage. For instance, the presence of *Lobularia maritima* attracts hoverflies, thereby enhancing their abundance and contributing to pest population reduction [13]. *Malva sylvestris* can serve as a valuable floral resource for hoverflies due to its high glycogen content, with its nectar playing a crucial role in promoting the proliferation of hoverfly populations [14,15]. Laboratory studies have demonstrated that predatory arthropods, such as ladybeetles and spiders, exhibit omnivorous feeding behaviors [16,17]. The inclusion of pollen and nectar significantly influenced their longevity, particularly in situations where the availability of prey was limited. The presence of *Perilla frutescens* flowers mixed with prey positively influenced the survival and early reproductive success of *Harmonia axyridis* [18,19]. In addition, members of the Miridae family also feed on pollen and nectar when prey is scarce; for example, it has been found that *Sesamum indicum* is ideally suited for planting on the edges of rice paddies to promote the biological control of rice pests by *Cyrtorhinus lividipennis* [20].

The Phytoseiidae family, commonly employed as biological control agents, comprises numerous omnivorous species that rely on both prey and floral resources, such as pollen and nectar, for their survival [21,22]. The oviposition of female adult mites has been found to be enhanced, likely due to the provision of pollen as a source of proteins and amino acids, thereby augmenting the fecundity of the predatory mites [23,24,25]. For instance, *Viburnum tinus* can provide pollen and oviposition sites for *Neoseiulus californicus*, thereby increasing the fecundity of this predatory mite [10]. When *N. californicus* consumed pistachio pollen, its fecundity was significantly enhanced [26]. Significant variations in the longevity and population parameters of predatory mites have been observed across different pollen types [27]. A number of scholars have conducted comparative analyses of the nutritional composition of pollen from *Prunus amygdalis*, *Zea mays*, and *Helianthus annuus*. They highlighted that variations in protein, lipid, and carbohydrate content among these pollens significantly influence the developmental timeline and reproductive capabilities of predatory mites [28]. Thus, the identification of suitable nectar and pollen plants is beneficial for the maintenance of predatory mite populations in agricultural fields.

The predatory mite *Neoseiulus bicaudus* (Acari: Phytoseiidae) has a wide distribution across several countries, encompassing Russia, Greece, Iran, Spain, Italy, Tajikistan, Cyprus, and the United States [29,30,31]. It is a facultative predator with generalist feeding habits, capable of regulating the populations of various pest species including spider mites, whiteflies, and thrips [32,33]. *N. bicaudus* has demonstrated effective control against low densities of *Tetranychus turkestani* on bean and cotton crops [34,35]. However, the prophylactic release of predatory mites is prone to escape or mortality due to insufficient food availability. Therefore, improving the resilience of released predatory mites is worth considering in order to achieve efficient and sustainable pest management [36].

Nectar and pollen plants can create suitable habitats for predatory mites, thereby enhancing their presence in agro-ecological environments and facilitating long-term pest control. For this study, we selected eight economically valuable nectar and pollen plant species that are cultivable in Xinjiang for screening purposes [37]. *Cnidium monnieri* (Umbelliferae) has a compound umbellate inflorescence that blooms from May to July. It has a distinctive fragrance, possesses ornamental properties, and can serve as a functional plant for corn, cotton, and other crops. Furthermore, it is a Chinese herbal medicine [38]. *Tagetes erecta* and *Centaurea cyanus* are herbaceous plants belonging to the Asteraceae family, having flowering periods from July to September and from April to May, respectively. The nectar they produce serves as a resource for crop pest predators and parasitic wasps. Additionally, both of these plants possess medicinal properties [39]. The annual herb *Borago officinalis* (Boraginaceae) exhibits cymes, blooming from May to October and yielding fruit from July to November. The flower possesses an ample amount of pollen, thus serving as a readily available protein source for insects [40]. Interestingly, *B. officinalis* also has a variety of edible, medicinal, ornamental, and cosmetic properties. In addition, the selection of *Brassica napus* and *L. maritima* (as plants in the family Brassicaceae) in this study is justified by their significant economic and ornamental value [41,42]. *Capsicum annuum* and *Medicago sativa*, which have the potential to serve as biological control agents, were selected as test plants [43,44]. *M. sativa* has a high nutritional value and is commonly used as the primary feed for livestock. *C. annuum* can be processed into food and spices, and is also used as a raw material in medicine, the chemical industry, and so on.

The availability of abundant pollen and nectar resources from nectar and pollen plants can positively influence the conservation of predatory mites. This phenomenon may preventively establish predatory mite populations before a target pest infests crops, thereby facilitating their colonization in the field and enhancing pest control efficiency. Consequently, this approach effectively mitigates the release of predatory mites, minimizes ecological risks, and addresses the limitations associated with conventional biological control [45]. Our objectives were to (1) assess the conservation role of nectar and pollen plants with respect to *N. bicaudus* and (2) evaluate the impact of pollen on *T. turkestani* predation by predatory mites.

## 2. Materials and Methods

### 2.1. Plant Cultures

Eight species of plants were tested: *C. monnieri* (Umbelliferae), *L. maritima* (Brassicaceae), *T. erecta* (Asteraceae), *C. cyanus* (Asteraceae), *B. officinalis* (Boraginaceae), *B. napus* (Brassicaceae), *C. annuum* (Solanaceae), and *M. sativa* (Leguminosae). These species were cultivated at the Experimental site of the Agricultural College of Shihezi University in Xinjiang Uygur Autonomous Region, China, in 2023 (E 85°57′, N 44°19′). Sowing took place on 18 April 2023 and watering was carried out fortnightly using drip irrigation tapes. No chemical pesticides were employed; only regular fertilizers and water management practices were implemented. Relevant plants producing nectar and pollen were utilized during their respective flowering periods. All plants bloomed from July through September 2023.

### 2.2. Mite Cultures

*T. turkestani* was from a laboratory colony. The colony was cultivated on cotton plants from the Experimental Station of Shihezi University. Green bean seedlings were planted in a pot first and the *T. turkestani* was assessed 10–15 days later. Transfer the *T. turkestani* to fresh beans when the bean seedlings have wilted and feed continuously for no less than 30 generations. All colonies were maintained in a growth chamber under controlled conditions of 26 ± 1 °C, 60 ± 5% relative humidity, and a 16:8 h (L:D) photoperiod.

*N. bicaudus* were collected from Ili, Xinjiang Uygur Autonomous Region, China, in 2013. They were reared on *Tyrophagus putrescentiae*. Bran (30% bran + 60% water + 10% yeast extract fermentation) was mixed in a box, then mixed with *T. putrescentiae*, followed by *N. bicaudus*, and reared for no less than 30 consecutive generations. The boxes were then placed in a growth chamber under controlled conditions of 30 ± 1 °C, 70 ± 10% relative humidity, and a 16:8 h (L:D) photoperiod for over 30 generations [46]. 

### 2.3. Effect of Different Nectar and Pollen Plants on Adult Longevity and Fecundity of Neoseiulus bicaudus

One newly hatched post-fertilized *N. bicaudus* female (a male mite having been provided for mating) was transferred onto the leaf surface in each chamber using a brush. The test chambers (5 cm × 5 cm × 0.3 cm) were constructed using three square transparent acrylic boards. A 2.5 cm diameter hole was cut in the center of the middle piece of the acrylic board, and the bean leaves were covered with a piece of filter paper that was longer than the chamber. Then, they were placed on the side of the holes and fixed to both sides of the chamber using clips [30]. In this experiment, the treatment group was set up by removing one flower of the selected plants (e.g., *C. monnieri* or *L. maritima*) with forceps. If the flowers were too large, the pistil was taken with forceps, and petals and leaves of the same size (1 cm × 1 cm) were cut using a scalpel (e.g., *T. erecta*, *C. cyanus*, *B. officinalis*, *B. napus*, *C. annuum*, and *M. sativa*). Only flower tissue was placed in the test chamber to provide food for predatory mites. The control group was set up—15 adult females reared on *T. turkestani*. The experimental chambers were placed in a growth chamber under controlled conditions; namely, temperature maintained at 26 ± 1 °C, relative humidity set at 60 ± 5%, and a 16:8 h (L:D) photoperiod. The survival and fecundity of *N. bicaudus* reared on different diets were recorded once every 24 h until the death of all individuals. The chambers were regularly replaced, and fresh food was provided on a daily basis. Each treatment was replicated over 30 times. 

### 2.4. Effect of Pollen from Nectar and Pollen Plants on the Predation Ability of Neoseiulus bicaudus

In this experiment, we reared predatory mites with a mixture of pollen from nectar and pollen plants and *T. turkestani*. The treatment groups were set up using newly hatched *N. bicaudus* feeding on *T. turkestani* + *C. cyanus* pollen, *T. turkestani* + *B. officinalis* pollen, and *T. turkestani* + *C. monnieri* pollen. The control group was *N. bicaudus* feeding on *T. turkestani* in the absence of pollen. The density of female adult spider mites was manipulated—5, 10, 15, 20, 25, and 35 per chamber in the *B. officinalis* and *C. monnieri* treatment groups for prey—with each density replicated ten times. To improve the fit of the predation function curves, the density of female adult mites was manipulated—5, 10, 15, 20, 25, 35, and 40 per chamber in *C. cyanus* treatment groups—with each density replicated ten times. At the same time, a sufficient amount (30 ± 10 mg) of the corresponding pollen was added to each chamber (the pollen used was obtained from flowers of nectar and pollen plants, collected by gently shaking them over clean petri dishes using a brush, and utilized on the same day of collection). This experiment required prior starvation of *N. bicaudus* in the test chamber. The experimental specimens were placed in a growth chamber at 26 ± 1 °C, 60 ± 5% relative humidity, and a 16:8 h (L:D) photoperiod. The *N. bicaudus* adults were kept with the prey for 24 h, after which the number of consumed *T. turkestani* individuals was quantified.

### 2.5. Statistical Analysis

The effects of different nectar and pollen plants on the longevity and fecundity of *N. bicaudus* were assessed by analyzing survival rate and fecundity data [47,48,49]. The raw data of the whole cohort were used to calculate the age stage-specific survival rate (*s_xj_*) (where *x* = age in days and *j* = stage), age stage-specific fecundity (*f_xj_*), age-specific survival rate (*l_x_*), age-specific maternity *(l_x_m_x_*), and age-specific fecundity (*m_x_*), according to the following equations:(1)$$l_{x}=\sum_{j=1}^{k}s_{\mathrm{xj}},$$
where *k* is the number of stages (1 age stage of female adult mites in this study);
(2)$$m_{x}=(\sum_{j=1}^{k}s_{\mathrm{xj}}f_{\mathrm{xj}})/(\sum_{j=1}^{k}s_{\mathrm{xj}})$$

The functional response of *N. bicaudus* was analyzed using the ‘frair’ package in the Rstudio 4.0I software [50]. Logistic regression was performed to assess the type of functional response. The equation used was
(3)$$\frac{N_{e}}{N_{0}}=\frac{\exp(p_{0}+p_{1}N_{0}+p_{2}N_{0}^{2}+P_{3}N_{0}^{3})}{1+\exp(p_{0}+p_{1}N_{0}+p_{2}N_{0}^{2}+P_{3}N_{0}^{3})},$$
where *N_e_* is the number of prey consumed and *N*_0_ represents the initial density of prey. The type of functional response was determined by the signs of the linear (*P*_1_) and quadratic (*P*_2_) coefficients. If the linear coefficient was significantly negative, the predator displayed a type II functional response. When the linear coefficient was positive and the quadratic coefficient was negative, the predator displayed a type III functional response.

The results of logistic regression indicated that the functional response of *N. bicaudus* was type II. Therefore, Rogers’ random predator model was used to fit the relationship between the number of prey consumed and initial prey density [51], using the following equation:(4)$$N_{e}=N_{0}[1-\exp(aN_{e}T_{h}-\mathrm{aT})],$$
where *a* represents the attack rate, *T* represents the total predation time (1 d), and *T_h_* represents the handling time. The attack rate (*a*) and handing time (*T_h_*) were obtained using the function “frair_fit” from the “frair” package in Rstudio. The significance of the attack rate (*a*) and handing time (*T_h_*) among different treatments was analyzed by employing “frair_compare” from the “frair” package in Rstudio. Handling time (*T_h_*) is the time spent attacking and consuming a prey before search is resumed [52]. The *a*/*T_h_* value reflects the feeding ability of predatory mites [53].

The influence coefficient of predation on *T. turkestani* in the presence of pollen was determined with reference to Manly’s preference index [54], calculated using the following equation:(5)$$\alpha =\frac{n_{\alpha }-n_{\beta }}{n_{\beta }}$$
where *α* represents the influence coefficient of predators, *n_α_* is the number of prey consumed in the presence of pollen, and *n_β_* is the number of prey consumed when there is no pollen. Negative values of the influence coefficient represent the negative effect of pollen on *T. turkestani* (decreasing), while positive values represent a positive effect (increasing comsumption). No effect was observed near zero. The larger the absolute value, the greater the effect.

## 3. Results

### 3.1. Effect of Different Nectar and Pollen Plants on Adult Longevity of Neoseiulus bicaudus

The age-specific survival rate (*l_x_*) provides a simplified overview of the survival rate (Figure 1). All *N. bicaudus* individuals exhibited 100% survival within 48 h after initiation, regardless of the (control or treatment) group. The *l_x_* curve of the control group declined on the 13th day. In addition, a higher *l_x_* of *N. bicaudus* was observed in the the *C. monnieri*, *L. maritima*, and *C. cyanus* treatment groups from 16 to 36 days, whereas a lower *l_x_* of *N. bicaudus* was observed under *T. erecta* and *C. annuum* treatments from 7 to 22 days.

Adult longevity under *C. monnieri* (24 days) was significantly longer than that of the control group (*F*_8,289_ = 18.712, *p* < 0.001) (Figure 2). There were no significant differences in the longevity under *L. maritima*, *C. cyanus*, *M. sativa*, *B. officinalis*, and *B. napus treatments*, compared with the control (*p* > 0.05). Longevity under *C. annuum* and *T. erecta* treatments exhibited a significant decrease compared to the control group (*p* < 0.05). The effect of *T. erecta* on the longevity of *N. bicaudus* (7 days) was lower than that of the other plants.

### 3.2. Effect of Different Nectar and Pollen Plants on Fecundity of Neoseiulus bicaudus

The age-specific fecundity (*m_x_*) and age-specific maternity (*l_x_m_x_*) of *N. bicaudus* exhibited variations across feeding on *Tetranychus turkestani* and the various nectar and pollen plants (Figure 3). *N. bicaudus* reared on *T. turkestani*, at an age of 4 days, exhibited the highest age-specific fecundity in female adults (*m_x_* = 1.9 eggs per female) and age-specific maternity (*l_x_m_x_* = 1.9 eggs per individual). *N. bicaudus* reared on *C. annuum* (*m_x_* = 1.45 eggs per female) and *B. officinalis* (*m_x_* = 1.4 eggs per female) diets exhibited significantly higher *m_x_* values in all treatments, while *N. bicaudus* reared on *L. maritima* (*m_x_* = 0.19 eggs per female) and *T. erecta* (*m_x_* = 0.05 eggs per female) diets exhibited the lowest *m_x_* values.

The fecundity serves as an indicator of the total number of eggs deposited per *N. bicaudus* female (Figure 4). The comparison of *N. bicaudus* fecundity under different treatments revealed a significantly higher fecundity, when reared on *T. turkestani*, compared to nectar and pollen plants (*F*_8,289_ = 47.168, *p* < 0.001). The fecundity under *C. monnieri* and *C. cyanus* treatments exceeded that of other plants, with 13.8 eggs and 13.4 eggs, respectively. In contrast, the impacts of *L. maritima* (2.1 eggs) and *T. erecta* (0.1 eggs) on the fecundity of *N. bicaudus* were lower compared to the control group and other plants.

The ANOVA results revealed the significant impact (*p* < 0.001) of pollen derived from diverse families and species on the oviposition rate and longevity of adult *N. bicaudus* females (Table 1).

### 3.3. Effect of Pollen from Nectar and Pollen Plants on the Predation Ability of Neoseiulus bicaudus

The predation of *Tetranychus turkestani* by *Neoseiulus bicaudus* varied with the density of prey in the presence of pollen (Table 2). For each treatment group and the control group, the consumption of *T. turkestani* gradually increased with increasing prey density. When the prey density was 5, 10, or 15 individuals/chamber, the results for all treatment groups were significantly lower than that of the control group (density 5: *F*_3,41_ = 4.478, *p* < 0.05; density 10: *F*_3,39_ = 7.024, *p* < 0.05; density 15: *F*_3,47_ = 4.729, *p* < 0.05). When the prey density reached 20 individuals/chamber, the predation rate of *N. bicaudus* was significantly lower than that of the control, only when pollen from *B. officinalis* and *C. cyanus* was added to the chamber (*F*_3,46_ = 11.049, *p* < 0.001). However, there was no significant difference between the control and treatment groups when the prey density was 25 or 35 individuals/chamber (density 25: *F*_3,42_ = 0.054, *p* > 0.05; density 35: *F*_3,47_ = 0.273, *p* > 0.05).

The logistic regression analysis revealed a significant negative linear coefficient *P*_1_ across all treatments (*p* < 0.001). This suggested that the predatory functional response of *N. bicaudus* to *T. turkestani* fits the type II response when provided with *C. cyanus* pollen, *B. officinalis* pollen, and *C. monnieri* pollen (Table 3). The predation amount of *N. bicaudus* gradually increased and subsequently reached a stable level in response to the rise in prey density (Figure 5).

The presence of pollen significantly decreased the attack rate (*a*) of *N. bicaudus* on *T. turkestani* compared to the absence of pollen (*B. officinalis* vs. Control: *z* = 3.5254, *p* < 0.05; *C. cyanus* vs. Control: *z* = 3.4145, *p* < 0.05; *C. monnieri* vs. Control: *z* = 2.8557, *p* < 0.05). Compared with the treatment groups after pollen addition, it was found that the female adult *N. bicaudus* had the highest attack rate (*a* = 2.41) on *T. turkestani* in the presence of pollen. The handing time (*T_h_*) of *N. bicaudus* without pollen (*T_h_* = 0.10 d) exhibited a longer duration, compared to that observed in the presence of pollen. The predation efficacy (*a*/*T_h_*) of *N. bicaudus* on *T. turkestani* was significantly higher than that with pollen (*a*/*T_h_* = 24.00), whereas the maximum consumption (1/*T_h_*) was lower than that in the presence of pollen (1/*T_h_* = 9.90 d^−1^) (see Table 3).

The influence coefficient (*α*) results demonstrated that the presence or absence of pollen had an effect on the predation of *T. turkestani* by *N. bicaudus* (Figure 6). The influence coefficient of predatory mites predation exhibited the highest absolute value in the presence of *C. cyanus* pollen, while it was observed to be the lowest in the presence of *B. officinalis* pollen. The overall value under *C. monnieri* treatment surpassed those for *B. officinalis* and *C. cyanus*. The value for *C. cyanus* surpassed that of *B. officinalis* at low densities, while they presented comparable values in later stages. Specifically, when the prey density was 5, 10, 15, or 20 individuals/chamber, the influence coefficients were all less than 0. When the prey density was 25 individuals/chamber, the influence coefficient was close to 0. However, when the prey density increased to 35 individuals/chamber, the value for *C. cyanus* exhibited a significant increase above 0, while the values for *B. officinalis* and *C. monnieri* approached 0.

## 4. Discussion

In this study, we evaluated the effects of eight different flowering plants on the longevity and reproduction rate of *N. bicaudus*. Additionally, we tested how their pollen influenced its predation ability.

Extending the lifespan of natural enemies enables their prolonged survival in the field, thereby contributing to the maintenance of natural enemy populations [55]. Our findings demonstrate that supplementing *N. bicaudus* with *C. monnieri* significantly enhanced its longevity, compared to only feeding it with *T. turkestani*, indicating the positive impact of *C. monnieri* in terms of sustaining *N. bicaudus* populations in agricultural environments. Similarly, *Passiflora edulis* prolonged the longevity of *Cotesia chilonis* and *Perilla frutescens* and had beneficial effects on the longevity of *Harmonia axyridis* [6,18]. On the contrary, *C. annuum* and *T. erecta* accelerated mortality in the predatory mites, suggesting that their utilization as nectar and pollen plants for biocontrol may not be suitable within agricultural environments.

The fecundity of *N. bicaudus* is significantly influenced by their ability to sustain a high reproductive capacity when provided with petals, pollen, nectar, and other tissues from flowering plants. This has crucial implications for evaluating the suitability of pollen and nectar plants. In this study, the production of *N. bicaudus* after the provision of flowering plants was as follows: *C. monnieri* > *C. cyanus* > *B. officinalis* > *M. sativa* > *B. napus* > *C. annuum* > *L. maritima* > *T. erecta*. The *C. monnieri* and *C. cyanus* treatment groups presented continuous egg-laying, indicating their sustained reproductive capacity, which is advantageous for the successful establishment of *N. bicaudus* in natural habitats. However, despite the provision of different flowering plants, the predatory mites exhibited a significantly reduced egg production compared to those fed *T. turkestani*. This phenomenon may be attributed to the necessity of a high-protein food source for successful reproduction, as well as the more favorable composition of the target prey in terms of reproductive potential. These findings are similar to those presented in previous research demonstrating that alyssum can serve as a resource subsidy for the anthocorid *Orius majusculus* during periods of prey scarcity in natural environments [56]. *Orius majusculus* exhibited continuous oviposition in the presence of *L. maritima* L.; however, maximum fecundity was only observed when prey were simultaneously present. Similarly, the total fecundity of the coccinellid *Stethorus gilvifrons* was not as high as in the presence of its natural prey, *T. turkestani*, when date palm pollen, maize pollen, or bee pollen were added to its diet [57,58].

In this study, we found that the family and species of nectar pollen plants were key factors influencing the longevity and fecundity of predatory mites. The observed variations in performance among different species were evidently significant. Among them, *C. monnieri*, *C. cyanus*, and *B. officinalis* exhibited superior qualities as nectar and pollen plants. Belonging to the Umbelliferae family, *C. monnieri* has umbrella-shaped, fragrant flowers and many nectaries, which effectively facilitate the conservation of diverse insects or predatory mites. *B. officinalis*, from the Boraginaceae family, is covered with white hairs for shelter and is high in nutrients, potentially explaining its attractiveness to predatory mites [59]. In the present study, it was found that *C. cyanus*, a member of the Asteraceae family, also enhanced the longevity and reproduction capacity of predatory mites. This phenomenon can be attributed to its high abundance of pollen and nectar, which are rich in essential nutrients such as sugars, fats, proteins, and flavonoids, among others [41]. The findings of this study showed that, although *T. erecta* belongs to the Asteraceae family, similar to *C. cyanus*, it led to poor survival and reproductive rates in *N. bicaudus*, possibly due to the presence of certain secondary compounds with adverse effects [60]. Similar observations were made with respect to *B. napus* and *L. maritima*, both of which are members of the Brassicaceae family. It is evident that plants from the same family but from different species have varying impacts on predatory mite populations. These results are consistent with previous research indicating that some flowering plant pollens are unsuitable for *Amblyseius swirskii* growth [26,61].

It has been reported that pollen can be used as a suitable food source for the long-term rearing of predatory mites [21,62]. It is of great significance to assess whether the presence of pollen interferes with the predation ability of predatory mites on pests. To further explore the impact of pollen on *N. bicaudus* cultivation, this study examined the functional response of *N. bicaudus* when preying upon *T. turkestani* in the presence of pollen [63]. The results showed that the predatory functional response of *N. bicaudus* to *T. turkestani* still fits the type II response when provided with *C. cyanus* pollen, *B. officinalis* pollen, and *C. monnieri* pollen, thereby indicating that the addition of pollen did not alter the feeding model [64,65]. This is consistent with a previous study [66], in which the presence of *Typha orientalis* pollen did not alter the functional response type of *Neoseiulus cucumeris*.

By analyzing the *a*/*T_h_* and 1/*T_h_* values of *N. bicaudus* in the presence and absence of pollen, it was found that the predatory efficiency of *N. bicaudus* on *T. turkestani* was highest when pollen was absent, while its maximum consumption was lowest. This result is consistent with the influence coefficient results, which were less than 0 (i.e., presence of pollen reduces predatory mite feeding on *T. turkestani*). The addition of pollen decreased the predatory ability of *N. bicaudus*, suggesting that *N. bicaudus* also feeds on a certain amount of pollen when pollen and prey coexist [67]. A similar phenomenon has been observed, in that providing pollen in mixed diets with high-quality prey resulted in a significant decrease in prey consumption [68]. We found that the presence of pollen had a negative impact on predation by pest mites at low prey densities; however, as the number of pest mites increased, the effect of pollen on predatory mite feeding decreased. When pests were available in large numbers, the predatory mites exhibited a preference for selecting pest mites, thus achieving optimal biological control effects. Moreover, when prey are scarce, pollen can serve as a supplementary food source to sustain the population of *N. bicaudus*. Similar findings have been observed when comparing the impact of plant and animal diets on various biological parameters of the phytoseiid mite, *Neoseiulus cucumeris* [69].

In summary, the presence of nectar and pollen plants, such as *C. monnieri*, *C. cyanus*, and *B. officinalis*, demonstrated positive effects on *N. bicaudus*, enhancing the life span and fecundity of this natural enemy species. Moreover, the preventive release of predatory mites to feed on food from nectar and pollen plants increases their survival during periods when prey is locally scarce. However, with an increase in the number of pests, the presence of pollen will not affect the effective control of predatory mites. At the same time, the considered nectar and pollen plants have aesthetic value, thus enhancing the agricultural environment. These findings establish a scientific foundation for incorporating nectar and pollen plants around target crops to effectively manage *T. turkestani* infestations.

## Figures and Tables

**Figure 1 insects-15-00190-f001:**
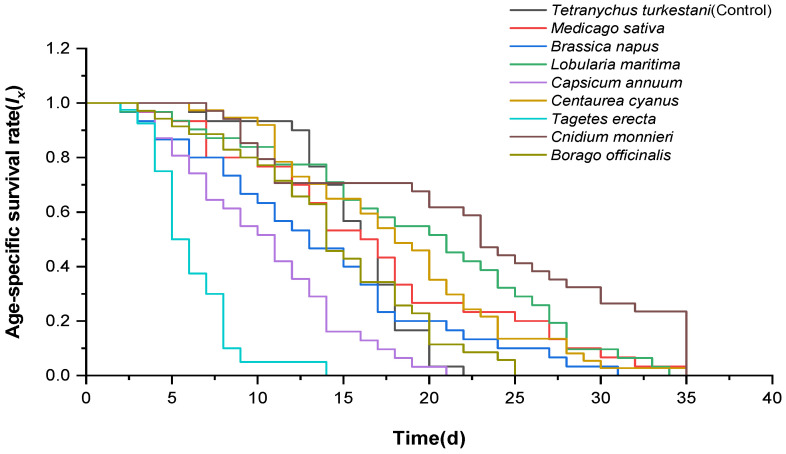
Age-specific survival rate (*l_x_*) of *Neoseiulus bicaudus* feeding on *Tetranychus turkestani* or nectar and pollen plants.

**Figure 2 insects-15-00190-f002:**
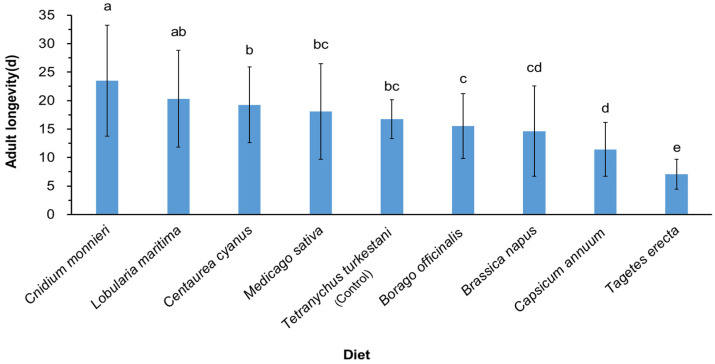
Adult longevity of *Neoseiulus bicaudus* feeding on different nectar and pollen plants. The values are means ± SE. Lowercase letters indicate significant differences among different treatments (*p* < 0.05). The data were analyzed by one-way ANOVA, SPSS 26.0.

**Figure 3 insects-15-00190-f003:**
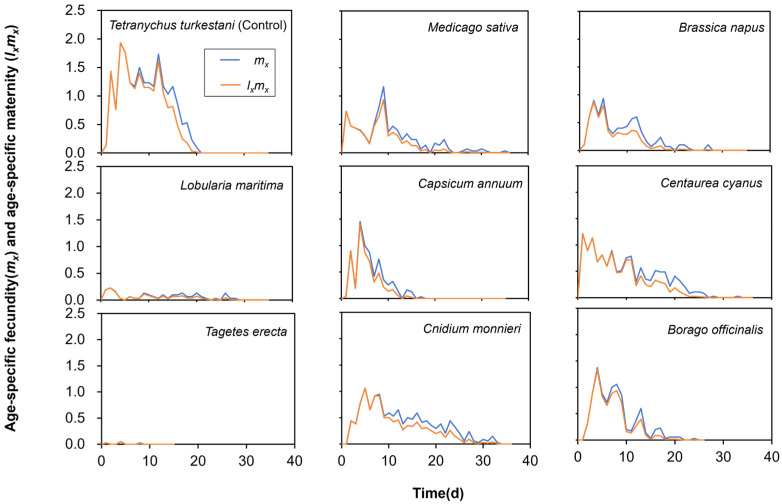
Age-specific fecundity of the total population (*m_x_*) and age-specific maternity (*l_x_m_x_*) of *Neoseiulus bicaudus* feeding on *Tetranychus turkestani* and nectar and pollen plants.

**Figure 4 insects-15-00190-f004:**
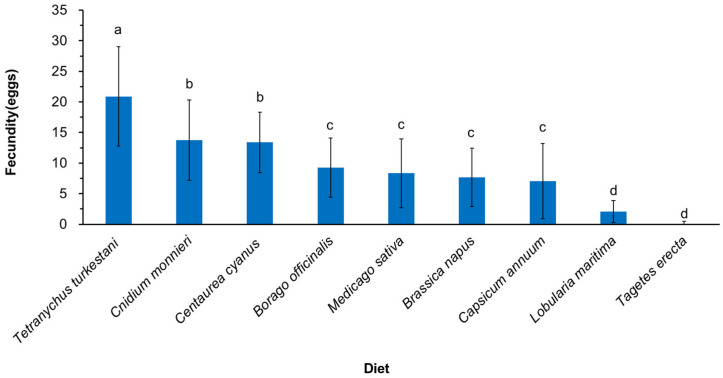
The fecundity of *Neoseiulus bicaudus* on different nectar and pollen plants. The values are means ± SE. Lowercase letters indicate significant differences among different treatments (*p* < 0.05). The data were analyzed by one-way ANOVA, SPSS 26.0.

**Figure 5 insects-15-00190-f005:**
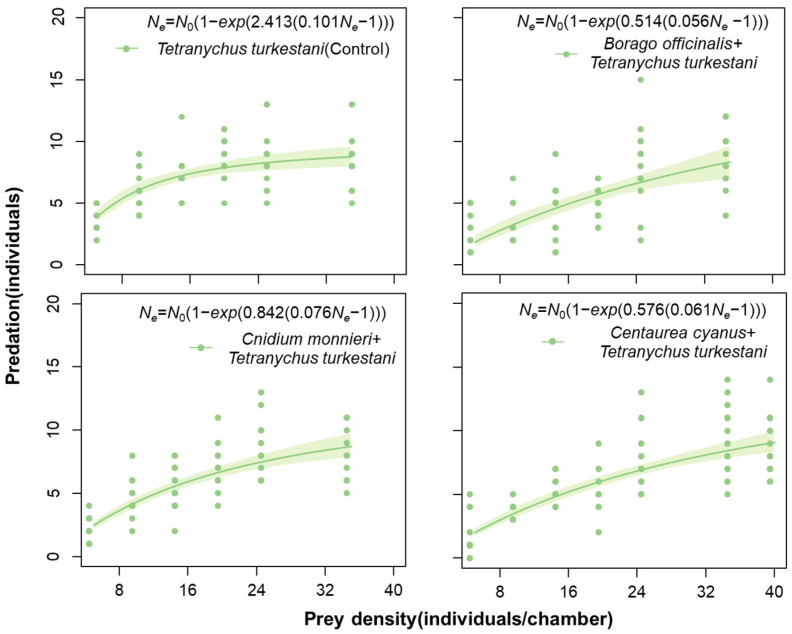
The predation of *Neoseiulus bicaudus* on *Tetranychus turkestani* in the presence of pollen. The shaded part in green are 95% confidence intervals.

**Figure 6 insects-15-00190-f006:**
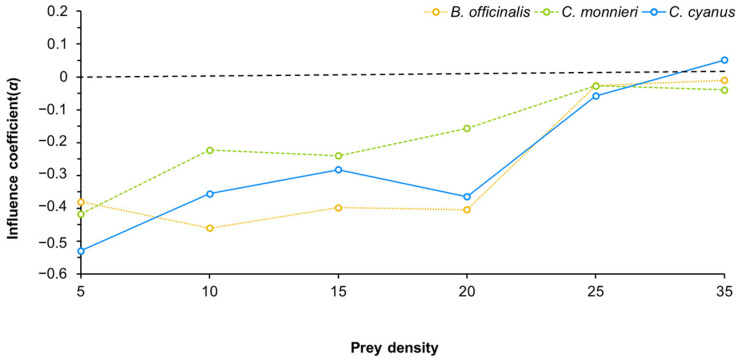
The influence coefficient of predation on *T. turkestani* by *N. bicaudus* in the presence of various pollens.

**Table 1 insects-15-00190-t001:** ANOVA results of oviposition and adult longevity of *Neoseiulus bicaudus* females in the presence of pollen from different families or species.

		Oviposition			Adult Longevity		
	Sources	df	*F*	*p*	df	*F*	*p*
Nectar and pollen plants	Family	5	18.048	<0.001	5	8.500	<0.001
	Species	7	35.475	<0.001	7	19.727	<0.001

**Table 2 insects-15-00190-t002:** Predation of *Neoseiulus bicaudus* on *Tetranychus turkestani* in the absence (control) or presence of different pollens. Data are means ± SE. Different capital letters indicate significant differences between the same treatments, different prey densities (*p* < 0.01); different lowercase letters indicate significant differences between the different treatments, same prey densities (*p*< 0.05).

Treatment	Prey Density (Individuals/Chamber)						
	5	10	15	20	25	35	40
*T. turkestani* (Control)	3.9 ± 1.04 Ca	5.9 ± 1.58 BCa	7.2 ± 1.99 ABa	8.3 ± 1.68 ABa	8.3 ± 2.10 ABa	8.5 ± 2.06 Aa	-
*T. turkestani + B. officinalis*	2.4 ± 1.50 Cb	3.2 ± 1.47 BCc	4.3 ± 2.29 BCb	4.9 ± 1.08 Bb	8.1 ± 3.35 Aa	8.4 ± 2.25 Aa	-
*T. turkestani + C. cyanus*	1.8 ± 1.57 Cb	3.8 ± 0.60 BCbc	5.2 ± 1.07 Bb	5.3 ± 1.81 Bb	7.8 ± 2.41 Aa	8.9 ± 2.51 Aa	8.6 ± 2.09 A
*T. turkestani + C. monnieri*	2.3 ± 0.96 Db	4.6 ± 1.50 Cb	5.5 ± 1.50 BCb	7.0 ± 1.86 ABa	8.1 ± 2.40 Aa	8.2 ± 2.11 Aa	-

**Table 3 insects-15-00190-t003:** Estimates of functional response of *Neoseiulus bicaudus* to female adult mites of *Tetranychus turkestani* in the presence of pollen. Data are mean ± SE. Lowercase letters indicate significant differences among different treatments (*p* < 0.05).

Treatment	Maximum Likelihood Estimate			Attack Rate	Handing Time	Predation Efficacy	Maximum Consumtion
	*P*_1_ ± SE	*z*	*p*	*a* ± SE	*T_h_* ± SE	(*a*/*T_h_*)	(1/*T_h_*)
*T. turkestani* (Control)	−0.0637 ± 0.0072	−8.901	<0.001	2.41 ± 0.53 a	0.10 ± 0.01 a	24.00	9.90
*T. turkestani + B. officinalis*	−0.0184 ± 0.0066	−2.795	<0.001	0.51 ± 0.09 b	0.06 ± 0.02 b	9.18	17.86
*T. turkestani + C. cyanus*	−0.0222 ± 0.0046	−4.786	<0.001	0.58 ± 0.09 b	0.06 ± 0.018 b	9.44	16.39
*T. turkestani + C. monnieri*	−0.0351 ± 0.0063	−5.527	<0.001	0.84 ± 0.14 b	0.08 ± 0.018 a	11.08	13.16

## Data Availability

The data presented in this study are available on reasonable request from the corresponding author.

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
