# Peer review of "Effects of Various Nectar and Pollen Plants on the Survival, Reproduction, and Predation of *Neoseiulus bicaudus"

_insects, 2024, doi:10.3390/insects15030190_

Round 1

Reviewer 1 Report

Comments and Suggestions for Authors

On line 50 in the introduction after resources you need a reference.

on line 81, after behaviors, you also need a reference there too.

On line 84 after in addition add "members of the"

On line 160 why the difference in RH?

On line 170 was it fertilized before or at hatching?  How and when did you determine the sex

on line 185 remove "had occurred" after individuals

Online 186 remove "was" after food

On line 256 change higher to longer

On line 255 AND 256 remove The and capitalize Adult

On line 259 remove the and capitalize longevity

on line 260 remove "when" after decrease

on line 264 remove the and capitalize adult

On line 280 change laid to deposited

On line 286 remove "when" after lower

On line 312 remove "of the treatments after all

On line 316 remove the s from pollen

On line 320 remove the after presence.  On line 320 what does handing time mean

On line 326 lower case female

On line 359 Remove The and capitalize fecundity and remove in female adult mites.  

On line 377 what is bee pollen

Author Response

Manuscript ID: insects-2865377

We sincerely thanks for your professional reviews that we have used to improve the quality of our manuscript. According to your nice suggestions, we have made corrections on our previous manuscript. The reviewer comments are laid out below in italicized font and specific concerns have been numbered. Our response is given in normal font and changes/additions to the manuscript are given in the red text. The detailed corrections are listed below.

Reviewer 1:

  1. On line 50 in the introduction after resources you need a reference.

We sincerely appreciate the valuable comments. We have checked the literature carefully and we have added reference to support this idea (reference: 1.Moretti, E.; Jones, C.; Schmidt-Jeffris, R. J. E.; Acarology, A., Alternative food sources for Amblydromella caudiglans (Phytoseiidae) and effects on predation. Experimental and Applied Acarology 2023, 89 (1), 29-44. http://dx.doi.org/10.1007/s10493-022-00767-y).

  1. On line 81, after behaviors, you also need a reference there too.

As your suggested, we have added literatures to this section as well (reference: 16. Schuldiner‐Harpaz T, Coll M. Estimating the effect of plant‐provided food supplements on pest consumption by omnivorous predators: lessons from two coccinellid beetles. Pest management science, 2017, 73(5): 976-983. 17. Eggs B, Sanders D. Herbivory in spiders: The importance of pollen for orb-weavers. PLoS One, 2013, 8(11): e82637.).

  1. On line 84 after in addition add "members of the".

   Thanks for your careful checks. Based on your comment, we have added “members of the” after in addition. (on line 87: In addition, members of the Miridae family also feed on pollen and nectar when prey is scarce).

  1. On line 160 why the difference in RH?

We sincerely thank you for careful reading. The reason for this question is that we have found more expansion at a relative humidity of 70 than 60 when we actually cultured populations of Neoseiulus bicaudus in the laboratory. This is beneficial for subsequent trials to progress.

  1. On line 170 was it fertilized before or at hatching?  How and when did you determine the sex.

   Thank you very much for your advice. In this experiment, the Neoseiulus bicaudus was fertilized after hatching. In the adult stage we can determine the sex by observing its morphology. ( this idea on line 181: One newly hatched post-fertilized N. bicaudus female (a male mite having been provided for mating) was transferred onto the leaf surface in each chamber using a brush).

  1. On line 185 remove "had occurred" after individuals.

    Thanks for your careful checks. Based on your comment, we have removed "had

occurred" after individuals (on line 200: The survival and fecundity of N. bicaudus reared on different diets were recorded once every 24 h until the death of all individuals).

  1. Online 186 remove "was" after food.

Thanks for your careful checks. Based on your comment, we have removed "was" after food (on line 201: The chambers were regularly replaced, and fresh food provided on a daily basis.).

  1. On line 256 change higher to longer.

     We think this is an excellent suggestion, we have changed higher to longer (on line 276: Adult longevity under C. monnieri (24 days) was significantly longer than that for the control group).

  1. On line 255 AND 256 remove The and capitalize Adult.

The revision was really helpful. In our resubmitted manuscript, we have removed The and capitalize Adult (on line 276 and 277: Adult longevity serves as an indicator of the survival of N. bicaudus (Figure 2). Adult longevity under C. monnieri (24 days) was significantly longer than that for the control group).

  1. On line 259 remove the and capitalize longevity.

    Thanks for your revision. In our resubmitted manuscript, we have removed the and capitalize longevity (on line 280: Longevity under C. annuum and T. erecta treatments exhibited a significant decrease, compared to the control group).

  1. On line 260 remove "when" after decrease

   Thank you for your valuable revision. We have removed "when" after decrease (on line 281: Longevity under C. annuum and T. erecta treatments exhibited a significant decrease, compared to the control group).

  1. On line 264 remove the and capitalize adult

   Thank you so much for helping me out with the revision. We have removed the and capitalize adult (on line 285: Adult longevity of feeding different nectar and pollen plants of Neoseiulus bicaudus).

  1. On line 280 change laid to deposited.

   We think this is an excellent suggestion. We have changed laid to deposited(on line 301: The fecundity serves as an indicator of the total number of eggs deposited per N. bicaudus female).

  1. On line 286 remove "when" after lower.

   The revision was a great help, and we really appreciate your assistance. We have removed "when" after lower. (on line 306: In contrast, the impacts of L. maritima (2.1 eggs) and T. erecta (0.1 eggs) on the fecundity of N.bicaudus were lower, compared to the control group and other plants.).

  1. On line 312 remove "of the treatments after all.

   Thanks for your careful checks. Based on your comment, we have removed "of the treatments after all(on line 335: This suggested that N.bicaudus displayed a Type II functional response to T. turkestani across all).

  1. On line 316 remove the s from pollen.

   We feel sorry for our carelessness. In our resubmitted manuscript, we have removed the s from pollen (on line 340: The predation of Neoseiulus bicaudus to Tetranychus turkestani in the presence of pollen).

  1. On line 320 remove the after presence.

   The revision was very helpful, and we truly appreciate your assistance. In our resubmitted manuscript, we have removed the after presence (on line 344: it was found that the female adult N. bicaudus had the highest attack rate (a = 2.41) on T. turkestani in the presence of pollen.).

  1. On line 320 what does handing time mean.

  Thank you very much for your question. Handling time is meant to be the time taken for each taker (from chasing to digestion), and in this study, the taker was Tetranychus turkestani. The unit of Th is d, I have added units in the text( on line 251-253: Handling time(Th) is meant to be the time taken for each taker (from chasing to digestion). The a/Th value reflects the control ability of natural enemies to pests).

  1. On line 326 lower case female.

  The revision was super helpful. In our resubmitted manuscript, we have lower-cased female (on line 351: Estimates of functional response parameter of Neoseiulus bicaudus to female adult mites of Tetranychus turkestani in the presence of pollen.).

  1. On line 359 Remove The and capitalize fecundity and remove in female adult mites.

  Thanks for your careful checks. Based on your comment, we have removed The and capitalize fecundity and remove in female adult mites (on line 383: Fecundity of N. bicaudus is significantly influenced by their ability to sustain high reproductive capacity when provided with petals, pollen, nectar, and other tissues from flowering plants).

  1. On line 377 what is bee pollen.

  Thank you very much for your question. Bee pollen is what I reviewed in the literature. It should be the pollen mass that bees bring back from honey harvesting (Reference 55: Ebrahimifar, J.; Shishehbor, P.; Rasekh, A.; Riddick, E. W., Effect of factitious diets on development and reproduction of the ladybird beetle Stethorus gilvifrons, a predator of tetranychid mites. Biocontrol 2020, 65 (6), 703-711. http://dx.doi.org/10.1007/s10526-020-10033-y).

  We tried our best to improve the manuscript and made some change marked in red in revised paper which will not influence the content and framework of the paper. We appreciate for Reviewers’ warm work earnestly, and hope the correction will meet with approval. Once again, thank you very much for your comments and suggestion.

Yours Sincerely,

Prof. Jie Su

Reviewer 2 Report

Comments and Suggestions for Authors

Review Report: insects-2865377

Neoseiulus bicaudus (Wainstein) has emerged as a potential biological solution for controlling spider mites and thrips, offering eco-friendly pest management possibilities. The significance of floral resources in providing habitat and nutrients to natural enemies, thereby improving their effectiveness in pest regulation, is well-established. This study aimed to evaluate how eight floral resources affect the longevity, fecundity, and predation ability of N. bicaudus.

The study highlights the intricate interactions between floral resources, predatory mite populations, and prey dynamics within agricultural ecosystems. By examining how different plant species impact N. bicaudus behavior and performance, this research contributes valuable insights to the development of sustainable pest management strategies. Implementing the recommended plant species has the potential to bolster predatory mite populations, leading to improved pest control outcomes while minimizing environmental harm.

Upon reviewing the manuscript, I acknowledge the effort put forth in conducting this research. However, after careful consideration based on the comments provided below, I recommend accepting this manuscript pending major revisions.

Comments for Authors:

The Materials and Methods section offers a thorough overview of the experimental setup and procedures. However, several deficiencies need addressing:

Lack of Specificity in Plant Cultures: While the study lists the eight plant species tested, it lacks detailed information regarding specific cultivars, planting techniques, and cultivation durations. Providing this data would enhance the study's reproducibility and clarity.

Inadequate Description of Mite Cultures: The section on mite cultures lacks specificity regarding maintenance protocols and conditions. Details such as colony maintenance frequency, colony sources, and rearing procedures are omitted. Inclusion of this information is essential for understanding the reliability and consistency of the mite populations used.

Experimental Design Clarity: Although the experimental procedures are outlined, there is ambiguity concerning the setup of treatment and control groups. Further clarification is needed on the establishment of treatment groups, particularly regarding the manipulation of floral resources and prey densities. A detailed step-by-step protocol would improve experiment reproducibility.

Addressing these shortcomings will enhance the Materials and Methods section, ultimately improving the manuscript's overall quality and reliability.

Author Response

Manuscript ID: insects-2865377

We sincerely thanks for your professional reviews that we have used to improve the quality of our manuscript. According to your nice suggestions, we have made corrections on our previous manuscript. The reviewer comments are laid out below in italicized font and specific concerns have been numbered. Our response is given in normal font and changes/additions to the manuscript are given in the red text. The detailed corrections are listed below.

Reviewer 2:

  1. Lack of Specificity in Plant Cultures: While the study lists the eight plant species tested, it lacks detailed information regarding specific cultivars, planting techniques, and cultivation durations. Providing this data would enhance the study's reproducibility and clarity.

   Thank you for your constructive comments. Because our flower seeds are purchased from the company, it is not possible to inquire about specific cultivars. Planting techniques and cultivation durations we have added in the resubmitted manuscript (on line 156-164: Choosing a plot with deep soil, loose structure and good drainage to plant the plants. Integrated planter for turning the ground, sowing seeds, setting up drip irrigation belts and laying mulch film. Sowing the seeds evenly to the soil surface according to the plant spacing of 33cm and row spacing of 66 cm, retaining 2 to 3 plants per hole. Sowing took place on April 18, 2023 and watering was done fortnightly using drip irrigation tapes. No chemical pesticides were employed; only regular fertilizers and water management practices were implemented. Relevant plants producing nectar and pollen were utilized during their respective flowering periods. All plants bloomed from July through September 2023.).

  1. Inadequate Description of Mite Cultures: The section on mite cultures lacks specificity regarding maintenance protocols and conditions. Details such as colony maintenance frequency, colony sources, and rearing procedures are omitted. Inclusion of this information is essential for understanding the reliability and consistency of the mite populations used.

   Thank you very much for your advice. We have added details such as colony maintenance frequency, colony sources, and rearing procedures in the resubmitted manuscript (on line 167-171: T. turkestani was from a laboratory colony. The colony were collected on cotton plants from the Experimental Station at Shihezi University. Plant the green bean seedlings first and access the T. turkestani 10-15 days later. Transfer the T. turkestani to fresh beans when the bean seedlings have wilted, feed continuously for not less than 30 generations; on line 174-176:Mixed bran (30% bran + 60% water + 10% yeast extract fermentation) in a box, then mixed with T. putrescentiae, followed by N. bicaudus, and reared for not less than 30 consecutive generations.).

  1. Experimental Design Clarity: Although the experimental procedures are outlined, there is ambiguity concerning the setup of treatment and control groups. Further clarification is needed on the establishment of treatment groups, particularly regarding the manipulation of floral resources and prey densities. A detailed step-by-step protocol would improve experiment reproducibility.

   Thank you very much for your advice. We have added details in the resubmitted manuscript (on line 181-183:One newly hatched post-fertilized N. bicaudus female (a male mite having been provided for mating) was transferred onto the leaf surface in each chamber using a brush; on line 191-194: Only flower tissue was placed in the test chamber to provide food for predatory mites. The control group was set up to provide 15 adult females reared on T. turkestani per day, no other food is provided; on line 200-202: The survival and fecundity of N. bicaudus reared on different diets were recorded once every 24 h until the death of all individuals. The chambers were regularly replaced, and fresh food provided on a daily basis; on line 205: In this experiment, we reared predatory mites with a mixture of pollen from nectar pollen plants and T. turkestani; on line 218, 219: This experiment required prior starvation of N. bicaudus in the test chamber.).

  We tried our best to improve the manuscript and made some change marked in red in revised paper which will not influence the content and framework of the paper. We appreciate for Reviewers’ warm work earnestly, and hope the correction will meet with approval. Once again, thank you very much for your comments and suggestion.

Yours Sincerely,

Prof. Jie Su

Round 2

Reviewer 2 Report

Comments and Suggestions for Authors

The authors have addressed all of my questions. I have no further remarks. I recommend accepting the manuscript in its current state.

Author Response

Dear Reviewers:

  Thank you very much for your recognition and support, we have benefited greatly from your suggestions and we will continue to improve our articles.

   Thank you and best regards.

   Yours Sincerely,

    Prof. Jie Su